# Scalability of Mach Number Effects on Noise Emitted by Side-by-Side Propellers

Caterina Poggi †, Giovanni Bernardini *,† , Massimo Gennaretti and Roberto Camussi

Department of Engineering, Università Roma Tre, 00146 Rome, Italy
* Correspondence: giovanni.bernardini@uniroma3.it
† These authors contributed equally to this work.

**Abstract:** This paper presents a numerical investigation of noise radiated by two side-by-side propellers, suitable for Distributed-Electric-Propulsion concepts. The focus is on the assessment of the variation of the effects of blade tip Mach number on the radiated noise for variations of the direction of rotation, hub relative position, and the relative phase angle between the propeller blades. The aerodynamic analysis is performed through a potential-flow-based boundary integral formulation, which is able to model severe body–wake interactions.The noise field is evaluated through a boundary-integral formulation for the solution of the Ffowcs Williams and Hawkings equation. The numerical investigation shows that: the blade tip Mach number strongly affects the magnitude and directivity of the radiated noise; the increase of the tip-clearance increases the spatial frequency of the noise directivity at the two analyzed tip Mach numbers for both co-rotating and counter-rotating configurations; for counter-rotating propellers, the relative phase angle between the propeller blades provides a decrease of the averaged emitted noise, regardless the tip Mach number. One of the main results achieved is the scalability with the blade tip Mach number of the influence on the emitted noise of the considered design parameters.

**Keywords:** aeroacoustics; distributed propulsion; numerical simulations

## 1. Introduction

The constant growth of urban overcrowding and pollution has made the individuation of alternative and environmentally sustainable solutions to the standard urban mobility a crucial issue. This has led to an ever-growing interest towards Urban Air Mobility (UAM), which represents an interesting solution for the infrastructure congestion, overcoming the limited capacity of ground transport. At the same time, the need to enhance the sustainability of conventional aircraft (namely for regional, continental and intercontinental transport) is also mandatory, in the perspective of a low impact next generation aviation. These two different applications share the common target to develop eco-friendly vehicles, in terms of both chemical and acoustic pollution, in response to the increasingly demanding requirements and certification rules. Focusing on the latter application, a great effort has been made to find new solutions to reduce the environmental impact of conventional, currently-employed aircraft. This approach has led to remarkable improvements but, in order to meet the targets in terms of performance and acoustic emissions set in [1], disruptive layouts and technologies have to be adopted.

In this scenario, the interest of manufacturers and researchers highly focused on the concept of Distributed Propulsion (DP) and, more specifically, Distributed Electric Propulsion (DEP). In these propulsive systems the thrust is delivered by multiple and decoupled propulsive devices [2]. Thanks to this peculiarity and the flexibility in the operating method of each device, this propulsive system is a viable green alternative to the traditional single/twin engine propulsive systems, mainly when electric-powered devices are exploited. Furthermore, this propulsive configuration lends itself well to be installed

on aircraft wing, usually resulting in an improvement of the system aerodynamics due to the lift increase produced by the propeller wakes/wing interactions [3,4]. Several research activities have been performed in this field and a lot of literature on this topic is already available, well testifying the growing interest in DEP layouts design. However, most of the existing literature mainly focuses on preliminary aircraft and propulsive system design or, at most, investigates aerodynamic issues [2,5–8]. Nevertheless, there are few studies aimed at assessing the aeroacoustic behaviour of such concepts (see, for instance [9,10]). A key point in DEP configurations design is the difficulty of accurately predicting the aeroacoustic effects due to the aerodynamic interactions occurring with other aircraft components, which may significantly modify both the aerodynamic performance and emitted noise of the system [11].

Focusing on aerodynamic interactions occurring between adjacent propellers a great amount of literature is already present, as for instance in [12] for tilt-rotor configurations and in [13–16] for drones. Furthermore, in [13], a numerical investigation of a quadcopter in hovering condition is proposed to examine the aeroacoustic effect of the rotors interactions, through a nonlinear vortex-lattice/vortex-particle method coupled with Farassat's formulation 1A. A similar investigation is presented in [14], where the effect of the relative position on the aerodynamics and acoustics of two side-by-side propellers has been investigated. In [17] a numerical/experimental investigation on the effect of hub separation distance, propeller tilting and blade phasing, for a two two-bladed co-rotating/counter-rotating propellers, has been presented. However, the available literature mainly concerns hovering conditions, whereas the aeroacoustic interaction effects in forward-flight remains relatively unexplored. Besides, although isolated multi-propeller systems or wing-mounted single rotors have been extensively investigated [18–22], installation effects on multi-rotor configurations have not been deeply examined so far. In particular, some studies regarding the aerodynamics of these configurations are available in the literature [3], whereas their acoustics remains relatively unexplored. Moreover, due to the novelty of this concepts, semi-empirical computationally efficient models are not present in literature and approximate solutions, as for instance, scattering models [23,24] are no longer applicable.

Driven by all these considerations, many European projects have started in the framework of Horizon 2020, among which VENUS (inVestigation of distributEd propulsion Noise and its mitigation through wind tUnnel experiments and numerical Simulations), which led to the present research, focused on the analysis of DEP systems aeroacoustics, through both numerical simulations and experimental activities. Regarding the latter, one issue to be carefully approached, is the sizing of the scaled model to be experimentally tested. To this purpose, some constraints have to be considered, as the wind tunnel geometrical characteristics, the power required to move the propeller and the permitted upper limit for the angular velocity to maintain an accurate control of the propeller blade shift. All these constraints limit the allowable experimental tip Mach number, which is significantly lower than that of the real model. Thus, the present paper provides an assessment of the tip Mach number effects on the aeroacoustic emissions, with the objective of defining general guidelines to suitably scale the model to be used in the experimental campaign. Specifically, the numerical analysis investigates the combined effects of the propeller tip Mach number, distance between hubs, direction of rotation and azimuth shift angle between blades. The ultimate goal is to assess the possibility of predicting the influence of the design parameters considered on the real configuration noise emission, starting from the experimental data related to the scaled model. As a by-product, the proposed analysis provides a systematic investigation of the combined effects of the blade tip Mach number and of the aforementioned geometrical/kinematic parameters on the noise emitted by multi-propeller configurations in forward flight, which, to the authors' knowledge, is not available in the current literature. The numerical investigation exploits for the aerodynamic analysis a velocity-potential based zeroth-order boundary element method that accounts for the effect of wake/body interaction [25] and, for the aeroacoustic one, an acoustic-pressure based

zeroth-order boundary element method, which solves the Ffowcs Williams and Hawkings equation [26].

The paper is structured as follows: first, in Section 2 the numerical solvers herein applied are briefly described, with Sections 2.1 and 2.2 outlining the aerodynamic and acoustic formulations, respectively. Then, Section 3 discuss the numerical investigation of a multi-rotor configuration to assess the effects of the propeller tip Mach number on system noise emissions when the rotor hubs distance, the azimuth phasing angles between the blades of adjacent propellers, and the direction of rotation change. Finally, Section 4 summarizes the main outcomes of the numerical analysis.

## 2. Numerical Tools

This section briefly describes the aerodynamic and aeroacoustic tools applied for the analysis of the multi-propeller configurations. All solvers herein used have been deeply checked in the recent past against available experimental data and numerical results for several rotorcraft in severe body-wake and wake-wake interaction conditions (see, for instance, ref. [27] for helicopter in BVI conditions and [28] for tilt-rotor configurations).

### 2.1. The Aerodynamic Tool

The aerodynamic solver exploits a Boundary Element Method (BEM)to solve the boundary integral formulation presented in [25]. Let us consider an incompressible, potential flow for which the velocity field, $v$, can be expressed as $v = \nabla \varphi$, where $\varphi$ denotes the velocity potential given by the sum of an incident potential, $\varphi_I$, and a scattered one, $\varphi_S$ (i.e., $\varphi = \varphi_I + \varphi_S$). The scattered potential is given by sources and doublets distributed over the body surface, $S_B$, and by the doublets distribution over the near wake, $S_W^N$, namely the portion of the wake close to the emitting trailing edge. The incident potential is instead generated by the doublets distribution over the far wake, $S_{W_F}$ (i.e., the near wake complementary wake regions) [25]. This splitting of the wake surface is such that only the far wake may impact bodies. The incident and scattered potentials are discontinuous across $S_W^F$ and $S_W^N$, respectively. Furthermore, the scattered potential equation is [25]

$$\varphi_S(\boldsymbol{x},t) = \int_{S_B} \left[ G(v_n - u_n) - \varphi_S \frac{\partial G}{\partial n} \right] dS(\boldsymbol{y}) - \int_{S_W^N} \Delta\varphi_S \frac{\partial G}{\partial n} \, dS(\boldsymbol{y}) \tag{1}$$

where $G = -1/4\pi \, ||\boldsymbol{y} - \boldsymbol{x}||$ is the 3D free-space Green function, whereas $\Delta\varphi_S$ is the jump of the scattered potential across the wake surface, which comes from the time history of scattered potential discontinuity at the corresponding body trailing edge exploiting Kutta's condition [29,30]. Furthermore, from the impermeability of the body surface $S_B$, $v_n = \boldsymbol{v}_B \cdot \boldsymbol{n}$, with $\boldsymbol{v}_B$ denoting the body velocity and $\boldsymbol{n}$ the unit normal vector to the body surface, oriented towards the fluid domain, while $u_n = \boldsymbol{u}_I \cdot \boldsymbol{n}$, where $\boldsymbol{u}_I$ is the far-wake induced velocity.

The numerical solution of Equation (1) is obtained through a zeroth-order boundary element method, which discretizes both the body and the near-wake surfaces into quadrilateral panels. Then, the scattered potential, $\varphi_S$, its normal derivative, $v_n$ and its jump across the wake, $\Delta\varphi_S$, are assumed uniformly distributed on each discretization panel (with the value matching that at the panel centroid), and the far-wake induced velocity, $\boldsymbol{u}_I$, is evaluated by exploiting the equivalence between the distribution of surface doublets and vortices, through the Biot-Savart law applied to vortices shaped as the wake panels contour. This formulation becomes singular in strong blade-wake interaction conditions (i.e., once the wake vortices impact a body). The Rankine vortex model is used to remove this singularity and ensure a regular induced velocity field even in severe blade-vortex interactions [25]. The wake shape evolution is obtained through a free-wake algorithm, which moves the vertices of the wake panels as per the velocity field induced by the bodies and their wakes. Equation (1) implies that the incident potential influences the scattered one through the induced-velocity, while the scattered potential influences the incident one through its trailing-edge jump that is transported by the wake material points and

defines the intensity of the far-wake vortices [25]. Once the scattered potential and the far-wake-induced velocity are known, the Bernoulli theorem allows the evaluation of the pressure field on the bodies [31] from whichthe body loads can be easily evaluated. Note that the Prandtl-Glauert transformation, applied to the pressure field, is used to include the compressibility effects to the aerodynamic formulation.

### 2.2. The Aeroacoustic Tool

The aeroacoustic analysis is based upon the boundary integral formulation for the solution of the Ffowcs Williams and Hawkings equation introduced in [24,32]. Specifically, the acoustic pressure field radiated by an impermeable body is given by

$$
E(\boldsymbol{x})p'(\boldsymbol{x},t) = -\int_{S_B} \rho_0 \left[ \boldsymbol{v} \cdot \boldsymbol{n} \boldsymbol{v} \cdot \nabla G + \left[ \boldsymbol{v} \cdot \boldsymbol{n}(1 - \boldsymbol{v} \cdot \nabla \theta) \right]^{\cdot} G \right]_\tau dS(\boldsymbol{y})
$$
$$
- \int_{S_B} \left[ (\boldsymbol{P}\boldsymbol{n}) \cdot \nabla G - (\dot{\boldsymbol{P}}\boldsymbol{n}) \cdot \nabla \theta G \right]_\tau dS(\boldsymbol{y})
\tag{2}
$$

where $p'$ is the pressure disturbance, $\boldsymbol{x}$ and $\boldsymbol{y}$ denote the observer and source positions, respectively, and $\boldsymbol{v}$ the local rigid blade velocity. In addition, $\rho_0$ is the density in the undisturbed medium, for inviscid flows $\boldsymbol{P} = [p - p_0]\boldsymbol{I}$ is the compressive stress tensor with $p_0$ denoting the pressure in the undisturbed medium. Furthermore, notation $[...]_\tau$ means that these quantities are evaluated at the emission time, $\tau = t - \theta$, with $\theta$ the time required by the signal emitted from $\boldsymbol{y} \in S_B$ to reach the observer at time $t$.

Also, in this case, Equation (2) is numerically solved through a zeroth-order boundary element method which discretizes the body into quadrilateral elements and assumes constant values (equal to their centroidal values) of the local surface velocity, $\boldsymbol{v}$, and acoustic pressure, $p'$, on each of them.

Note that, as demonstrated in [26], Equation (2) is fully equivalent to Farassat 1A formulation [33], with the first and second integrals corresponding to the thickness and the loading contributions, respectively.

## 3. Numerical Results

The numerical investigation herein proposed is aimed at assessing the effect of the propeller blade tip Mach number, $M_t$, on the noise emitted by two adjacent co-rotating and counter-rotating propellers in forward-flight conditions. The examined configuration consists of two side-by-side, five-bladed propellers, with parallel rotational axes and diameter $d = 0.67$ m. The chord length of the untapered blades is equal to 0.025 m. For the aerodynamic analysis, each blade surface is discretized by 840 panels, with 30 panels in the chordwise direction (15 on the upper side and 15 on the lower side of the airfoil) and 28 panels spanwise, whereas 5040 panels are used to discretise the corresponding two-revolution length free wake surface (28 panels in the radial direction and 180 panels along the azimuth direction).

The time discretisation consists of 360 time-steps per rotor revolution, corresponding to 1°-azimuth resolution. Aeroacoustic calculations are performed using the same blade mesh and time step used for the aerodynamic analysis. These discretization parameters are obtained as the best compromise between accuracy and computational efficiency, chosen as the result of a convergence analysis on the rotor thrust performed by gradually increasing their values.

The investigation concerns the effect of the propeller blade Mach number on the near-field and far-field noise emitted by the propeller-wing system when the relative distance and phase between the propeller blades change. Two angular velocities are examined, 5000 rpm and 6730 rpm, corresponding to $M_t = 0.52$ and $M_t = 0.7$, respectively. Note that the first velocity is typical of scaled models for wind-tunnel tests, whereas the second one is consistent with standard operative conditions of real-life DEP configurations. The effects of the distance between the propeller hubs and of the propeller blade phasing are investigated considering three tip-clearances, $TC_1 = 0.05d$, $TC_2 = 0.125d$, and $TC_3 = d$ and three phase

shifts ($12°, 24°$, and $36°$). All configurations are tested by keeping unchanged both the advance ratio (increasing the freestream velocity from 25 m/s to 33.65 m/s as $M_t$ increases) and the thrust (equal to 189 N for each propeller), in that the proposed investigation aims at studying the noise provided by similarly loaded propellers. A preliminary trim analysis is performed for each configuration to identify the collective angle providing the required thrust.

The results of the numerical investigations are shown in terms of Overall Sound Pressure Levels (OASPLs, evaluated on the basis of the first nine BPFs) on a sphere with radius $R = 40d$ for the far-field investigation and $R = 5d$ for the near-field analysis, centered at the midpoint of the two propeller hubs. Specifically, two views of the sphere are shown. Considering a right-handed frame of reference with origin at the center of the sphere, the $z$-axis coincident with the advancing direction and pointing forward and the $y$-axis directed starboard (see Figure 1), the northern view (NV) consists of the view of the sphere as seen from the positive $x$-axis, whereas in the southern view (SV) it is observed from the negative $x$-axis.

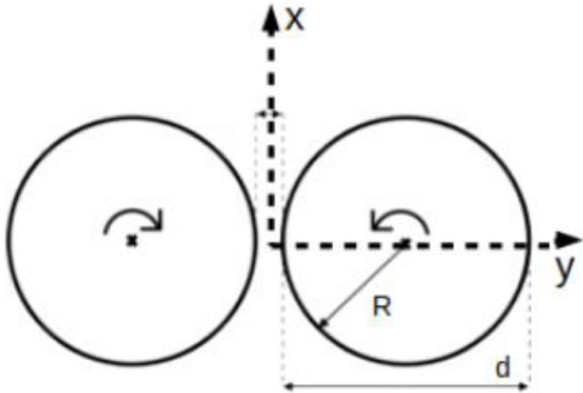

**Figure 1.** Sketch of the examined configurations, counter-rotating propellers.

First, the effect of the blade tip Mach number on the far-field noise produced by counter-rotating propellers is evaluated for the defined values of TCs. The corresponding predicted OASPL are compared in Figure 2.

These results show that, although an OASPL magnitude difference of about 10 dB is observed to be induced by the Mach number change analyzed, the directivities remain similar for all TCs, especially in the region closer to the rotor disk. In particular, for the lower Mach number conditions, the northern view presents a symmetric distribution of the noise radiated by the two propellers with respect to the $xz$-plane and three lobes with high OASPL values that collapse into a single high-intensity region located close to the $yz$-plane as the tip-clearance increases. A similar behavior is also observed for the higher Mach number conditions although, in this case, two trilobate structures are present, with the one below the $yz$-plane much more intense than that above it. Instead, in the southern view, a more wavy noise distribution is observed, generated by constructive or destructive interference between sound fields emitted by the two propellers. The corresponding spatial frequency increases with the tip clearance and the noise waves remain in the same azimuth positions regardless the tip Mach number. Some differences appear in the region upstream the rotor disk, where the radiated noise is quite uniformly distributed for $M_t = 0.52$, whereas wave interference patterns are clearly visible for $M_t = 0.7$.

For a clearer interpretation of the noise distribution, Figure 3 shows the differences between the OASPL given by different TCs, for both blade tip Mach numbers considered. In this figure and in the following ones, $\Delta OASPL_{ij}$ is introduced to denote the difference between OASPLs evaluated for tip-clearance $TC_i$ and tip-clearance $TC_j$.

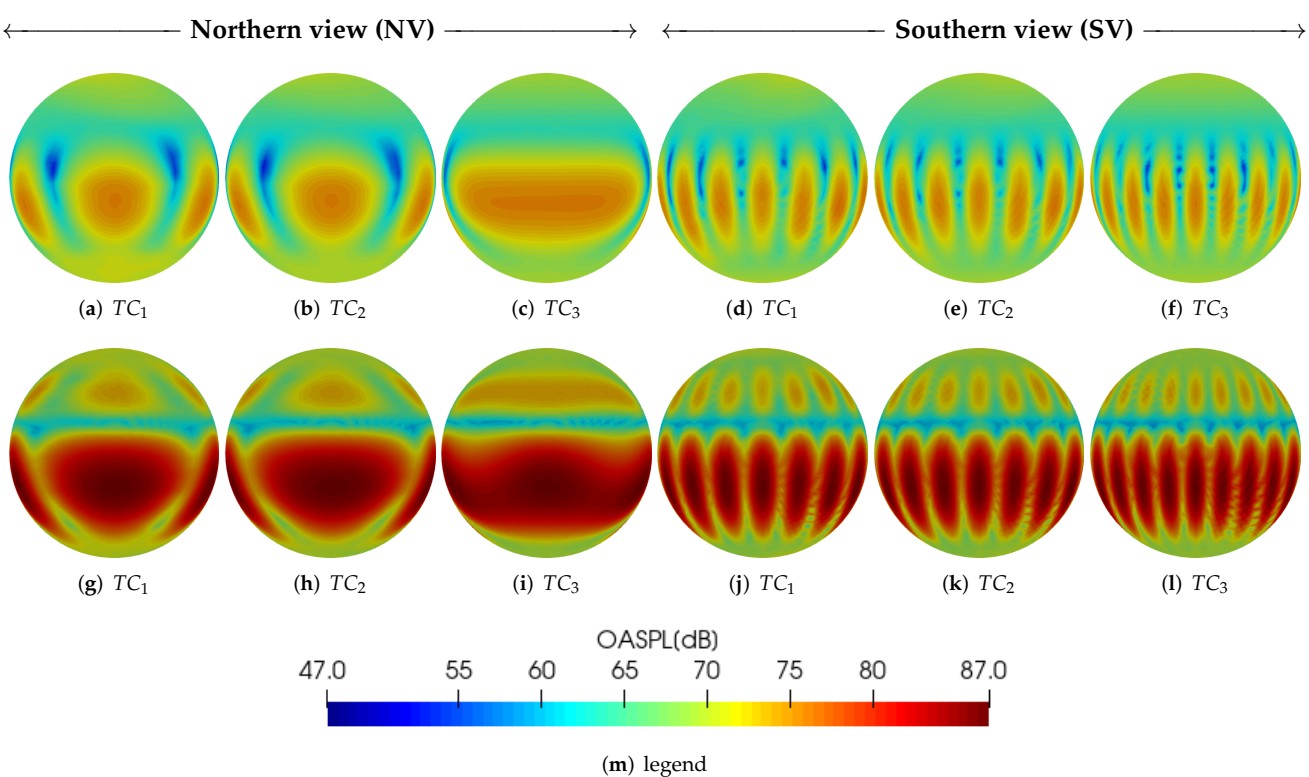

**Figure 2.** Counter-rotating propellers, influence of TC on OASPL, for $M_t = 0.52$ (upper) and $M_t = 0.7$ (lower).

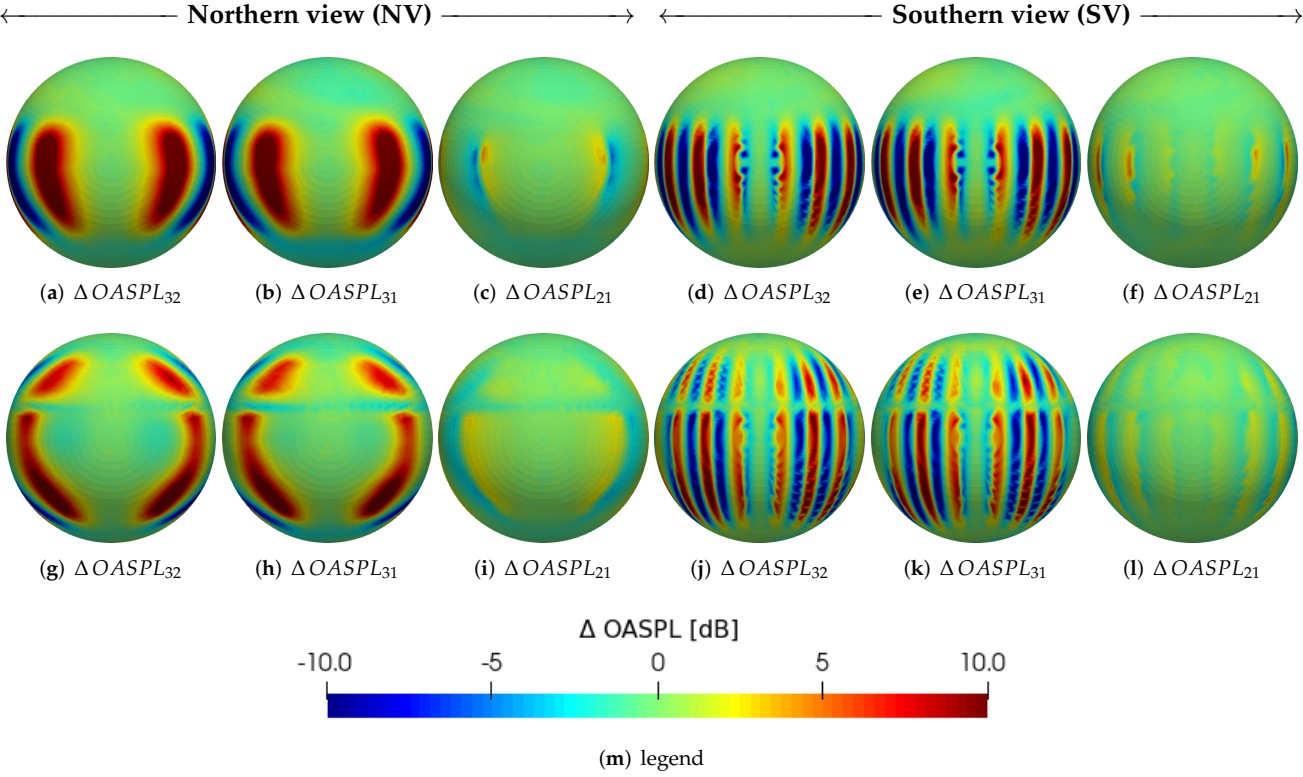

**Figure 3.** Counter-rotating configuration, OASPL difference vs. TC difference, for $M_t = 0.52$ (upper) and $M_t = 0.7$ (lower).

As expected, the most significant difference occurs between the configurations with the higher and lower tip clearances. At $M_t = 0.52$ the differences are localized near the

rotor disk plane, whereas upstream the radiated noise is almost independent of the tip clearance. At $M_t = 0.7$ the $\Delta$ OASPL is not negligible also in the microphones near the polar region upstream the propeller disk. Furthermore, the increase of the spatial frequency for increasing tip clearance observed in Figure 2 generates $\Delta$ OASPL interference patterns in the southern view that are similar for both tip Mach numbers.

Next, the same analyses are performed for the configuration with co-rotating propellers. The corresponding evaluated OASPL distributions are shown in Figure 4. It is worth noting that, in this case, the SV and the NV coincide due to the geometric symmetry of this configuration with respect to the $xy$-plane.

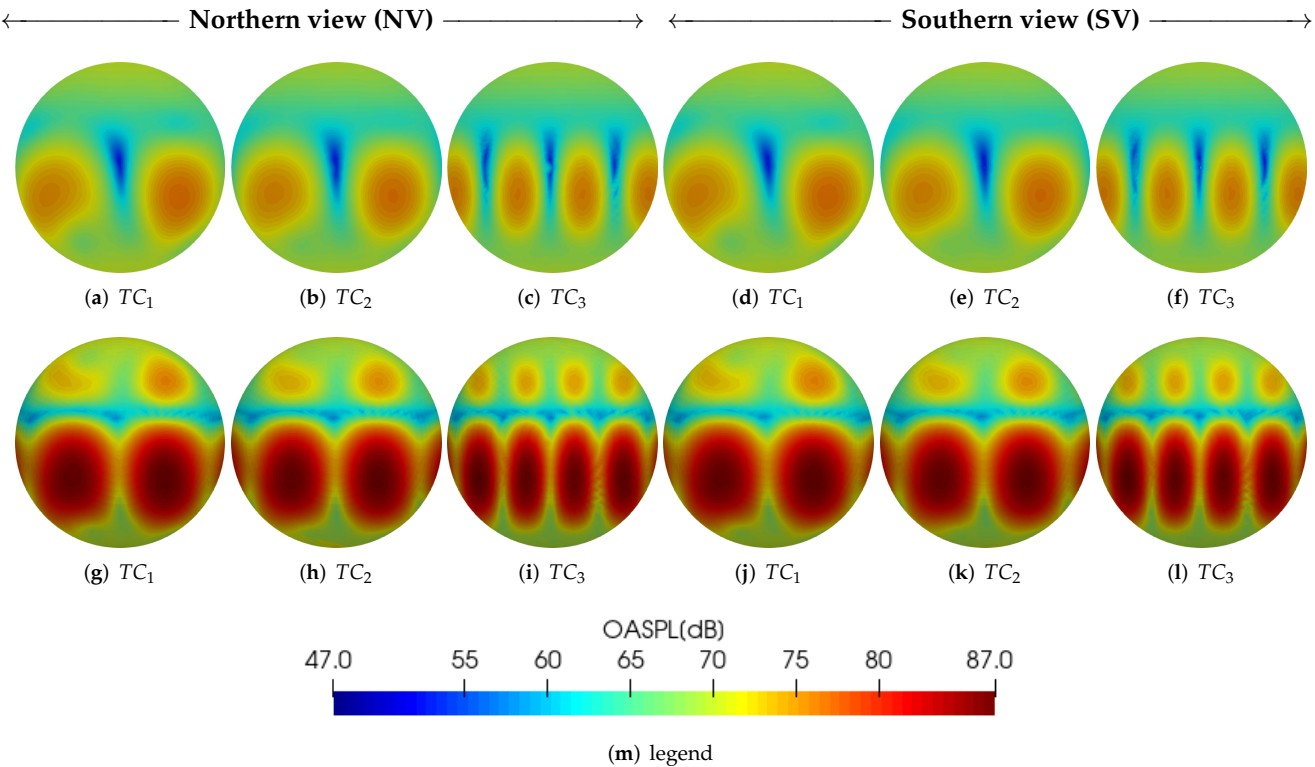

**Figure 4.** Co-rotating propellers, influence of TC on OASPL, for $M_t = 0.52$ (upper) and $M_t = 0.7$ (lower).

Akin to the counter-rotating propeller analysis, in order to get a better insight about the tip-clearance effect on the emitted noise, Figure 5 shows the differences between the OASPL given by different TCs, for both blade tip Mach numbers considered.

These results confirm the general trend observed also for the counter-rotating case: greater differences are present between maximum and minimum tip-clearances, the increase of the spatial frequency for increasing tip clearance observed in Figure 2 generates $\Delta$ OASPL interference patterns are similar for both tip Mach numbers.

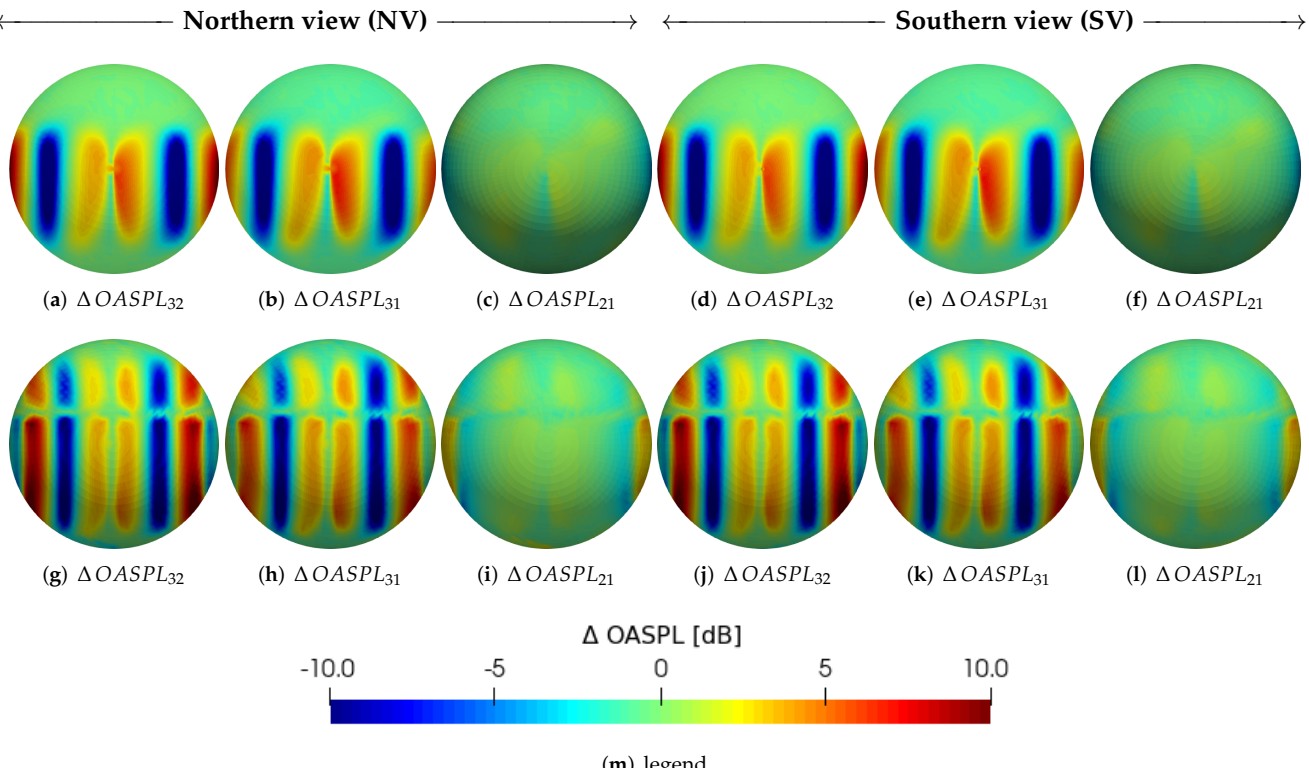

$$\longleftarrow\text{---- } \textbf{Northern view (NV)}\text{ ----}\longrightarrow \longleftarrow\text{---- } \textbf{Southern view (SV)}\text{ ----}\longrightarrow$$

(**a**) $\Delta\,OASPL_{32}$   (**b**) $\Delta\,OASPL_{31}$   (**c**) $\Delta\,OASPL_{21}$   (**d**) $\Delta\,OASPL_{32}$   (**e**) $\Delta\,OASPL_{31}$   (**f**) $\Delta\,OASPL_{21}$

(**g**) $\Delta\,OASPL_{32}$   (**h**) $\Delta\,OASPL_{31}$   (**i**) $\Delta\,OASPL_{21}$   (**j**) $\Delta\,OASPL_{32}$   (**k**) $\Delta\,OASPL_{31}$   (**l**) $\Delta\,OASPL_{21}$

Δ OASPL [dB]

-10.0     -5     0     5     10.0

(**m**) legend

**Figure 5.** Co-rotating configuration, OASPL difference vs. TC difference, for $M_t = 0.52$ (upper) and $M_t = 0.7$ (lower).

The second numerical investigation carried out regards the effect of the phase shift, $\gamma$, on the emitted noise. The analysis is performed for both co-rotating and counter-rotating configurations, for both Mach numbers, for all the propeller hub distances considered. Specifically, Figures 6 and 7 show, respectively for the counter-rotating and co-rotating configurations, and for increasing tip-clearance, the far-field OASPL directivity patterns on a circular array of observers with radius equal to $40d$ and centered at the midpoint of the dual propeller system, lying on the rotor disk plane. Note that, for $\psi = 0°$ the observer is placed at $(x = -R, y = 0)$, and that positive $\psi$ corresponds to positive rotation around the $z$-axis.

For all the configurations examined, the effect of the phase angle between the blades of the two propellers is a shift of the azimuth positions where the signals from the two propellers generate constructive or destructive interference. Furthermore, Figure 6 shows that, for the counter-rotating configurations, $M_t = 0.52$ and $TC = 0.125d$, a significant reduction of noise (up to 25 dB) is achieved for $\gamma = 36°$ and $120° \le \psi \le 240°$ (in the same azimuth region, the radiated noise presents a visible reduction also for $\gamma = 24°$). A similar behavior is also observed for $M_t = 0.7$, and for all the tip-clearances, although the reductions are much smaller and appearing in a narrower azimuth range. The results for co-rotating propellers presented in Figure 7 show that the phase angle $\gamma = 36°$ produces a noise magnitude reduction of about 10 dB for $TC = d$ and azimuth ranges $60° \le \psi \le 120°$ and $240° \le \psi \le 300°$. For all the other cases, the noise increases in some directions and decreases in others.

Thus, at least for the configurations herein investigated, the strategy of reducing the emitted noise by a proper choice of the relative phase angle between propellers seems to be more effective for the counter-rotating configuration than for the co-rotating one.

This conclusion is confirmed by the examination of the values of the averaged OASPL computed over the sphere that are reported in Table 1. However, further considerations can be drawn from Table 1.

**Table 1.** Averaged OASPL.

| | $M_t$ | Tip-Clearance | $\gamma = 0°$ | $\gamma = 12°$ | $\gamma = 24°$ | $\gamma = 36°$ |
|---|---|---|---|---|---|---|
| | | 0.05$d$ | 68.78 dB | 68.51 dB | 67.90 dB | 66.97 dB |
| | 0.52 | 0.125$d$ | 68.67 dB | 68.43 dB | 67.80 dB | 66.97 dB |
| | | 1$d$ | 69.09 dB | 68.78 dB | 67.67 dB | 65.23 dB |
| Counter-rotating | | 0.05d | 75.07 dB | 74.74 dB | 74.05 dB | 73.49 dB |
| | 0.7 | 0.125$d$ | 75.00 dB | 74.67 dB | 73.96 dB | 73.35 dB |
| | | 1$d$ | 75.36 dB | 74.84 dB | 73.64 dB | 72.78 dB |
| | | 0.05$d$ | 67.31 dB | 67.97 dB | 68.46 dB | 68.65 dB |
| | 0.52 | 0.125$d$ | 66.92 dB | 67.74 dB | 68.46 dB | 68.70 dB |
| Co-rotating | | 1$d$ | 67.45 dB | 67.96 dB | 68.12 dB | 68.00 dB |
| | | 0.05$d$ | 73.38 dB | 74.04 dB | 74.84 dB | 75.14 dB |
| | 0.7 | 0.125$d$ | 73.24 dB | 73.83 dB | 74.77 dB | 75.14 dB |
| | | 1$d$ | 73.62 dB | 73.95 dB | 74.45 dB | 74.65 dB |

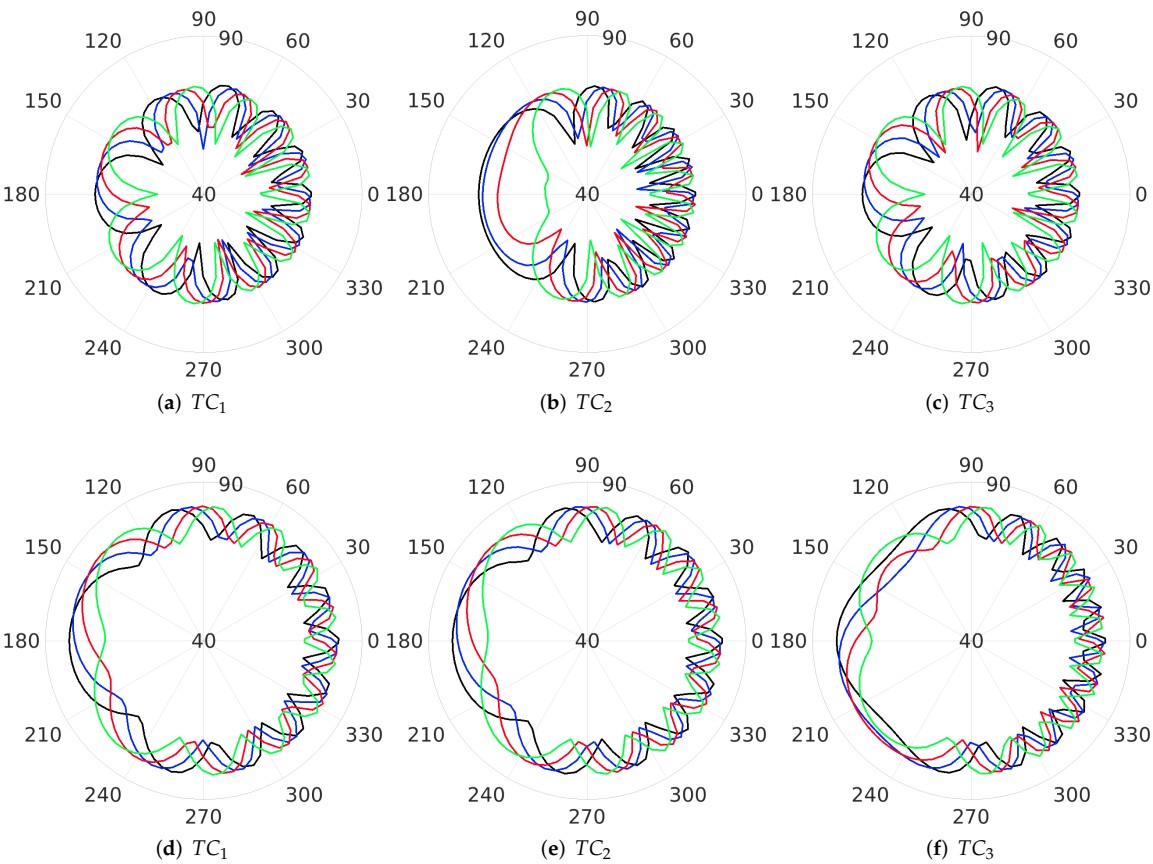

**Figure 6.** Counter-rotating propellers, OASPL directivity pattern on rotor disk for different TCs and different phase shifts (black line $\gamma = 0°$, blue line $\gamma = 12°$, red line $\gamma = 24°$, green line $\gamma = 36°$); upper $M_t = 0.52$, lower $M_t = 0.7$.

First, regardless of the propeller direction of rotation and the tip Mach number, for $\gamma = 0°$, the highest averaged OASPL is obtained for the maximum TC, whereas the minimum averaged OASPL is obtained for TC = 0.125$d$. This behavior can be the result of a combination of source-observer distance (inversely proportional to the tip-clearance) and unsteady propeller loading (directly proportional to the tip-clearance).

Second, regardless of the tip Mach number and tip clearance, increasing the relative phase angle yields a noise increase for the co-rotating configuration and a noise decrease

for the counter-rotating one (thus confirming the overall trend observed in Figures 6 and 7). For the counter-rotating configuration, the maximum relative phase angle seems to be the most effective in reducing the noise at both Mach numbers.

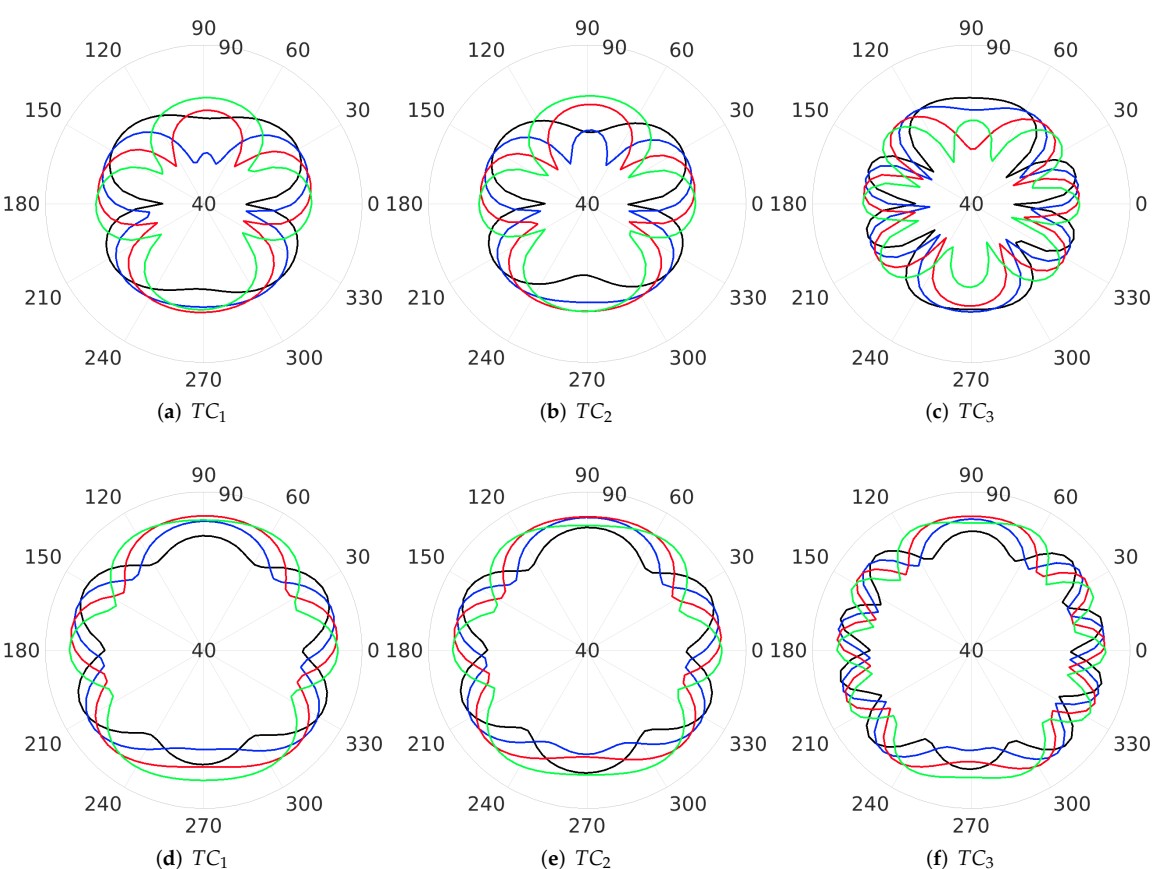

**Figure 7.** Co-rotating propellers, OASPL directivity pattern on rotor disk for different TCs and different phase shifts (black line $\gamma = 0°$, blue line $\gamma = 12°$, red line $\gamma = 24°$, green line $\gamma = 36°$); upper $M_t = 0.52$, lower $M_t = 0.7$.

Finally, the above investigations are also performed for near-field observers (located on the sphere of radius $5d$). First, for the counter-rotating configuration, Figure 8 shows the difference between the OASPL evaluated in the far-field hemisphere (radius $40d$) and the near-field one (radius $5d$), indicated as $\Delta OASPL$, for the defined TCs and Mach numbers. Note that for this comparison, a suitable scaling of the OASPLs in terms of the ratio between the sphere radii is performed.

In this case, the differences between near-field and far-field noise distribution are limited to a few dBs in wide areas of the northern hemisphere, especially in regions close to the rotor disk plane. Greater discrepancies arise near the poles (where the noise emitted is less significant) and in the southern hemisphere. From these results it is inferred that the dipole contribution to the noise is negligible also on the hemisphere of radius $5d$. This could be due to the phase shifts of the dipole signals coming from the two propellers of the counter-rotating configuration, which produce a destructive interference even at small observation distances.

Figure 9 shows the same results for the co-rotating configuration. In this case, the differences between near- and far-field predictions are greater, although the overall trend is the same as the counter-rotating configuration, in terms of both tip-clearance and tip Mach number effects. In this case, the phase shifts of the dipole signals seem to be such that the dipole contributions do not cancel out each other in the near field.

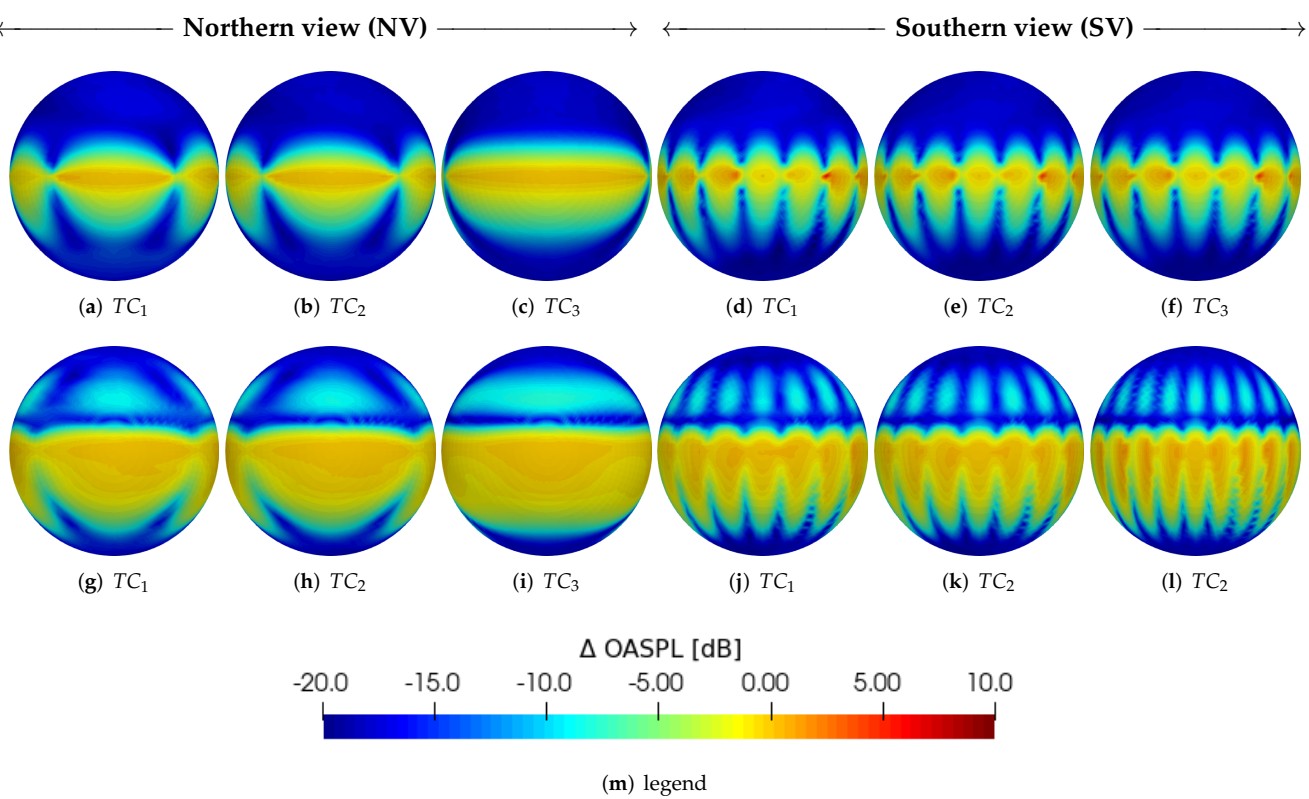

**Figure 8.** Counter-rotating configuration, difference between far-field and near-field OASPL for the defined TCs and for $M_t = 0.52$ (upper) and $M_t = 0.7$ (lower).

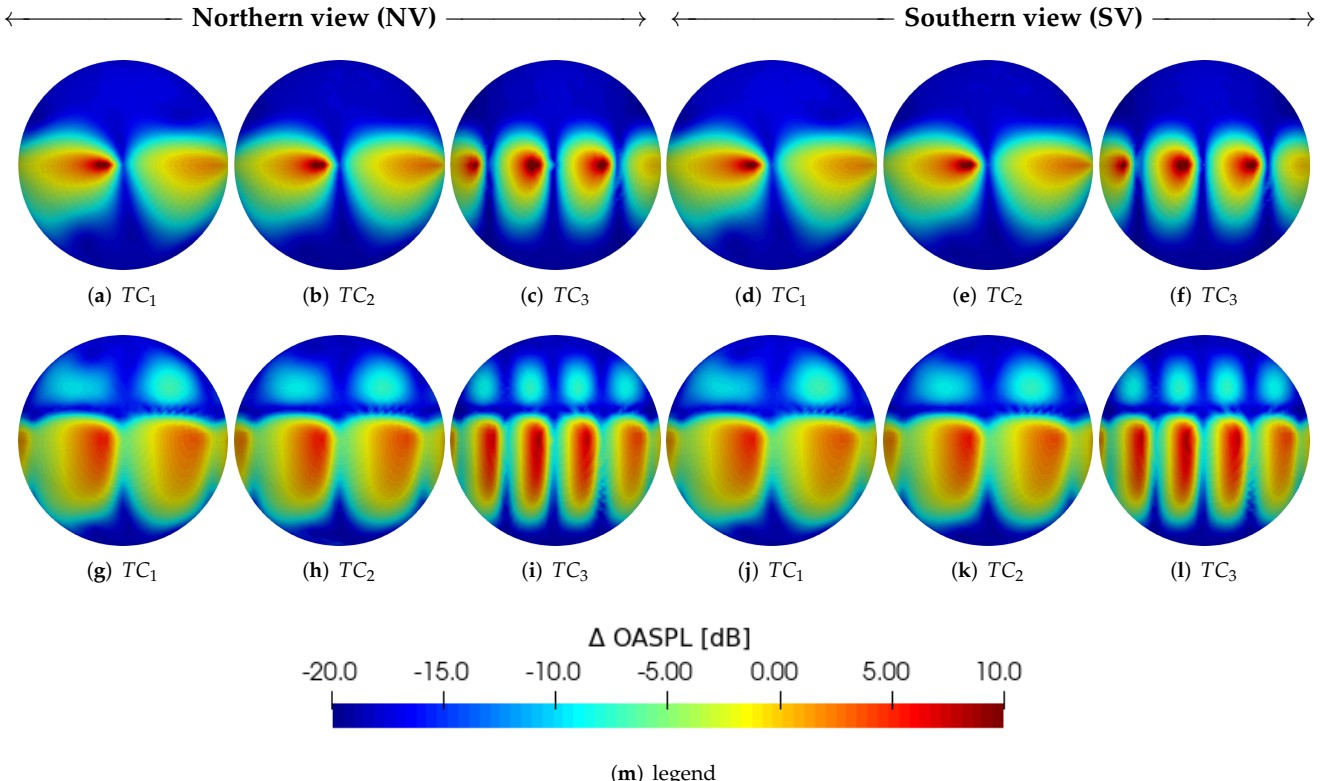

**Figure 9.** Co-rotating configuration, difference between far-field and near-field OASPL for the defined TCs and for $M_t = 0.52$ (upper) and $M_t = 0.7$ (lower).

Finally, Figures 10 and 11 show, respectively for the counter-rotating and co-rotating case, the comparison between near-field and far-field directivity patterns evaluated on the circular array of microphones located on the rotor disk plane, for different phase shifts and tip Mach numbers. For the sake of conciseness, only results for the phase angles $\gamma = 0°$ and $\gamma = 36°$ and TC $= 0.05d$ and TC $= d$ are shown (the conclusions that are drawn in these cases are valid also for all the other configurations examined). Note that, also in this case, for comparison purposes, the OASPLs have been scaled by the ratio between the hemisphere radii.

These results prove that the directivity patterns of the near-field and far-field predictions are in excellent agreement for the counter-rotating case, whereas some discrepancies appear in the co-rotating configuration. Indeed, in this latter case an angular shift between them is present, due to the non-vanishing dipole contribution in the near-field noise discussed above.

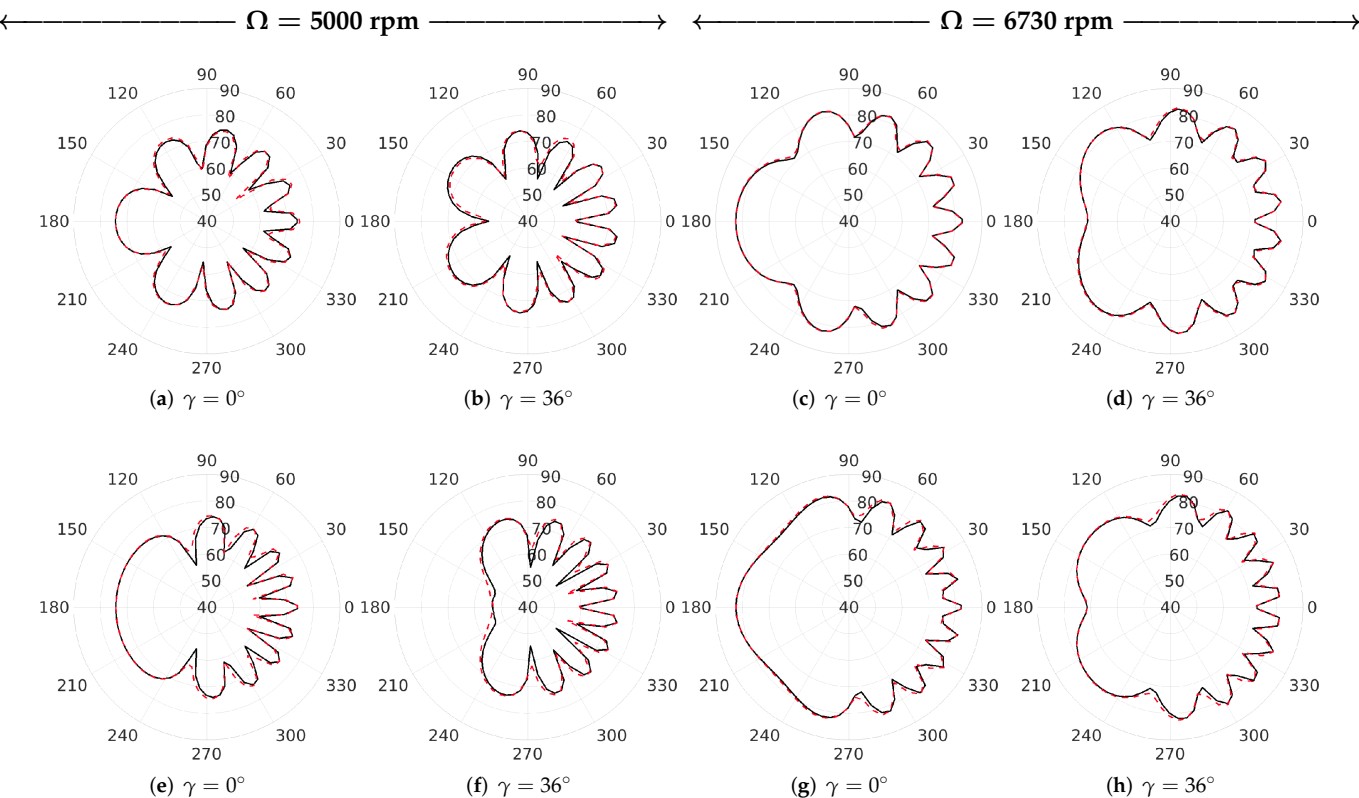

**Figure 10.** Counter-rotating configuration, comparison between far-field and near-field OASPL directivity patterns; red line near field, black line far-field; TC $= 0.05d$ (upper), TC $= d$ (lower).

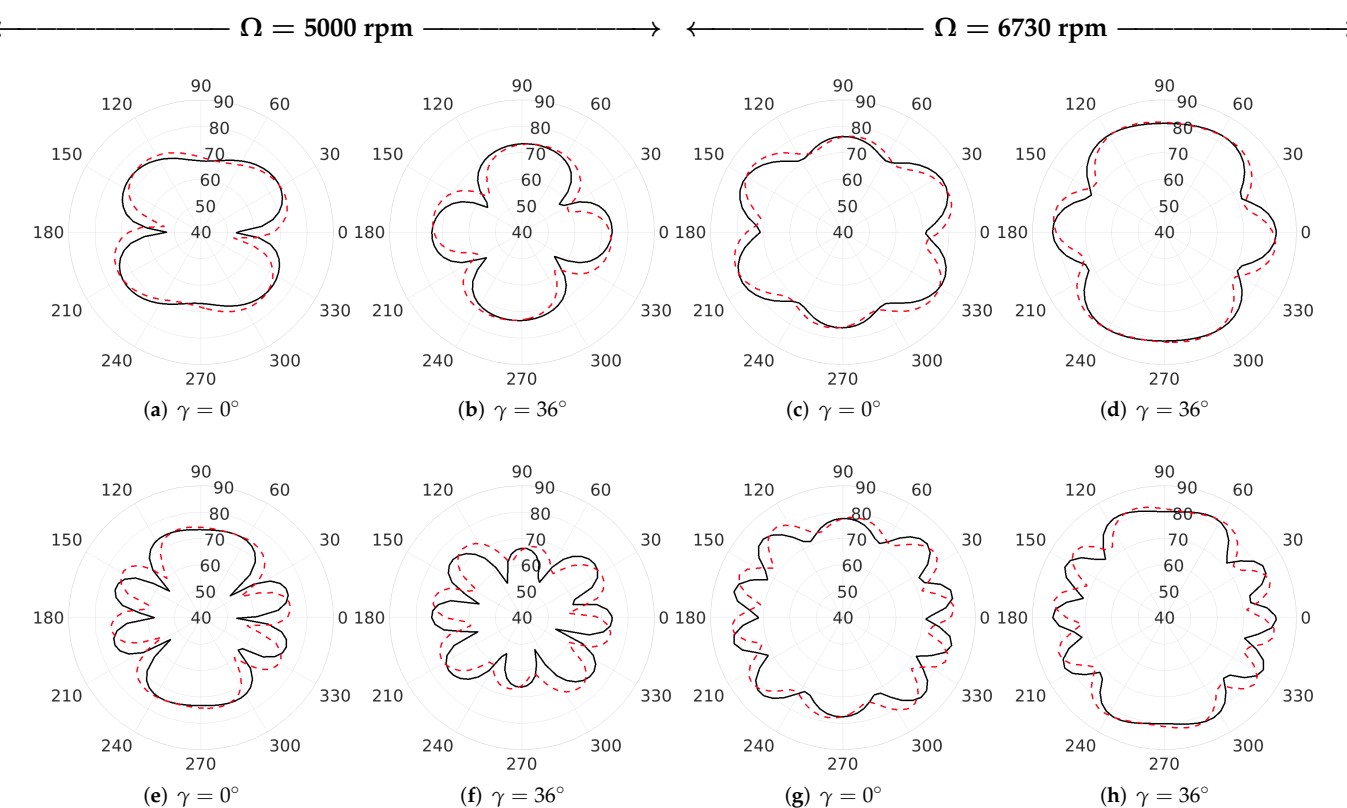

**Figure 11.** Co-rotating configuration, comparison between far-field and near-field OASPL directivity patterns; red line near field, black line far-field; TC = 0.05*d* (upper), TC = *d* (lower).

## 4. Conclusions

This paper presents a numerical investigation on the effect of the blade tip Mach number on the noise emitted by two side-by-side propellers, in co-rotating and counter-rotating configurations. In particular, the proposed analyses investigate the influence of the propeller blades' tip Mach number on the emitted noise sensitivity to design parameters like the relative distance and the relative phase angle between the propeller blades. Both near-field and far-field noise propagation are examined. From the numerical investigations discussed some considerations can be drawn:

- for both co-rotating and counter-rotating propellers, the effects on the emitted noise of the tip-clearance is similar for both tip Mach numbers, with those related to the lower tip Mach number mainly localized near the rotor disk plane;
- for the counter-rotating propellers configuration the relative phase angle alters the noise directivity pattern on the rotor disk plane, generally providing an angular shift of it; however, for specific values of phase angle and tip-clearance, a significant reduction of the emitted noise magnitude is observed for a wide angular range, particularly for the low Mach number flight condition;
- for the co-rotating propellers configuration the relative phase angle produces angular shift of the noise directivity pattern on the rotor disk plane, but slight modifications of the magnitude; anyway, these effects are more evident for the lower tip Mach number;
- the differences between near-field and far-field noise predictions decrease with the tip Mach number, increase with the tip clearance, and are greater for the co-rotating propellers configuration; this occurs because, in this case, the dipole-type propagation is still non-vanishing in the near field (whereas it is almost negligible in the counter-rotating propellers configuration because of a mutual cancellation effect).

A consequence of the results achieved is that, in the perspective of an experimental investigation, even if a low-Mach-number model is tested for near field noise measurements,

these could be suitably scaled to obtain reliable predictions of the far-field noise propagation of the real, higher-Mach-number configuration.

**Author Contributions:** Conceptualization, C.P., G.B., M.G. and R.C.; methodology, C.P., G.B. and M.G.; software, C.P., G.B. and M.G.; investigation, C.P.; data curation, C.P.; writing—original draft preparation, C.P. and G.B.; writing—review and editing, C.P., G.B., M.G. and R.C.; visualization, C.P.; supervision, G.B., M.G. and R.C. All authors have read and agreed to the published version of the manuscript.

**Funding:** This work has been partially supported by the European Union Horizon 2020 research and innovation programme under project VENUS (inVestigation of distributEd propulsion Noise and its mitigation through wind tUnnel experiments and numerical Simulations), grant agreement No 886019.

**Conflicts of Interest:** The authors declare no conflict of interest.

## Abbreviations

The following abbreviations are used in this manuscript:

| | |
|---|---|
| DEP | Distributed Electric Propulsion |
| DP | Distributed Propulsion |
| OASPL | Overall Sound Pressure Level |
| NV | Northern View |
| SV | Sounthern View |

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
