# Peer review of "Scalability of Mach Number Effects on Noise Emitted by Side-by-Side Propellers"

_applsci, doi:10.3390/app12199507_

Round 1

Reviewer 1 Report

A minor revision is need before This paper can be received and published on Applied Sciences, the comments are as follows:

1 What is the definition of Δ OASPL shown in Fig,3?

2 The layout of the titles in Figures should be improved.

3 Is the calculated results need to be verified with experiment results? 

Reviewer 2 Report

The manuscript reports a numerical work about the scalability of Mach number effects on noise emitted by side-by-side propellers due to the different Mach numbers operated in the experiments and actual applications. The authors adopt BEM based on incompressible, potential flows and the Farassat 1A formulation of Ffowcs Williams and Hawkings equation to evaluate the acoustic emission of the propellers. 

While the work is interesting and has significant contributions for the Distributed Electric Propulsion, the manuscript suffers some major shortcomings, so I cannot recommend their publications. Details are as follows.

  1. Missing mesh and surface information

In the manuscript, the information about the chosen surface and the meshes are all missed. Where and how are the surfaces chosen? What are meshes adopted for the integration? It is important for readers to know these technical details of the modelling.  

  1. Missing grid convergence study

Grid convergence study is an important step to show the credibility of the models. It must be included in the manuscripts. 

  1. Missing parts in methodology

In Line 143, how the surface velocity is evaluated? Could the authors please describe some of its details? It is because this velocity is essential in evaluating the noise emission.
